# Miracidia as main source for autofluorescence of *Schistosoma mansoni* eggs

**Danielle Segóvia Chrysóstomo de Almeida Pereira[1], Laila Oliveira Vaz Oliveira[1], Felipe Tonon Firmino[1], Thomas Hanscheid[2], Rock Pulak[3], Malcolm Jones[4], Silvio Dolabella[5], Deborah Negrão-Corrêa[6], Carlos Graeff-Teixeira[1]/+**

[1]Universidade Federal do Espírito Santo, Núcleo de Doenças Infecciosas, Helmintologia, Vitória, ES, Brasil
[2]Universidade de Lisboa, Escola de Medicina, Lisboa, Portugal
[3]Union Biometric, Holliston, MA, United States of America
[4]University of Queensland, School of Veterinary Sciences, Gatton, Australia
[5]Universidade Federal de Sergipe, Centro de Ciências Biológicas e da Saúde, Entomologia e Parasitologia Tropical, Aracaju, SE, Brasil
[6]Universidade Federal de Minas Gerais, Instituto de Ciências Biológicas, Departamento de Parasitologia, Belo Horizonte, MG, Brasil

**BACKGROUND** Egg detection still has a role in schistosomiasis control, as a screening strategy or to provide a reference standard for the assessment of the accuracy of other diagnostic tools. The Helmintex method is highly sensitive but laborious, and several improvements of it, including automated egg detection, are currently under development.

**OBJECTIVE** We conducted a preliminary evaluation of *Schistosoma mansoni* eggs' autofluorescence as a distinctive marker amid very complex fecal sediments.

**METHODS** Eggs from mouse livers and human feces were examined under a fluorescence microscope.

**FINDINGS** More intense green fluorescence (greater for miracidia than for eggshell) was consistently detected using a B-2A filter (FITC, 420-495 nm).

**MAIN CONCLUSIONS** These findings may help to improve diagnostic methods, especially with automated egg detection systems. Besides access to safe water and adequate sanitation, as well as health education and the treatment of infected individuals, laboratory diagnosis is a key measure that can help eliminate schistosomiasis as a public health problem.

Key words: schistosomiasis - autofluorescence - coproparasitology - Helmintex - egg

Although the sensitivity limitations of egg detection–based methods for the diagnosis of parasitic infections are well known, the Kato-Katz fecal thick smear is still used widely in many endemic areas for the diagnosis of schistosomiasis. Due to eggs' peculiar size and morphology,[1] their identification in fecal samples enables highly specific classification, what is valuable for the establishment of reliable reference data for the assessment of other diagnostic tools' accuracy. The Helmintex (HTX) method involves the use of magnetism to isolate *Schistosoma mansoni* eggs from feces; it has 100% sensitivity for egg burdens > 1.3 eggs per gram.[2] Despite its accuracy, this method is labor intensive, and several improvements are currently underway. One promising possibility is a detection step using a flow system coupled with morphometric and fluorescent detectors. We thus investigated the autofluorescence of *S. mansoni* eggs to aid their identification in ongoing flow system detector development.

## MATERIALS AND METHODS

*Schistosoma mansoni* eggs were obtained from (i) experimentally infected BALB/c mouse livers (ML) (CEUA-UFMG ethical approval 41/2024) and (ii) sediment pools from naturally infected human feces (HF) from a biorepository under proper ethical clearance (CEP-CCS-UFES CAAE 71027323.7.0000.5060). Eggs isolated from ML after seven-eight weeks post infection were kept in 10% formalin solution,[3] and those isolated from HF were fixed in 50% alcohol and stored at 10ºC.

Eggs from MLs (*n* = 20) were inoculated into 30 µL uninfected HF sediment produced by the HTX method.[4] Briefly, the HTX steps were sequential passage through 500-, 150- and 45-µm sieves and centrifugation with ethyl-acetate in water. The sediment with eggs was spread onto microscope slides and left to dry at room temperature. The same procedure was performed with three sets containing 12 to 14 HF eggs. The slides were

Financial support: FAPES-DECIT (grant 2021-HGDKG - Pesquisa para o SUS; grant PROFIX 2022-1DNS6), CNPq (grant and fellowship 304070/2023-8; grant 404583/2023-7).

+ Corresponding author: graeff.teixeira@gmail.com | ● https://orcid.org/0000-0003-2725-0061

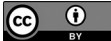

examined under a fluorescence microscope (Eclipse 80i, LED light source; Nikon Corporation, Tokyo, Japan) with the filter sets indicated in Fig. 1.

## RESULTS

All eggs from MLs and HF exhibited visible autofluorescence under the B-2A filter, with the miracidium having the most intense signal and the eggshell having a weaker signal (Fig. 2). Less-intense fluorescence was observed with the other filters (Fig. 1). These findings suggest that *S. mansoni* egg autofluorescence originates primarily from functional molecules in the miracidium, rather than from structural components such as the eggshell.

## DISCUSSION

Autofluorescence is a natural optical property observed in the tissues and cellular organelles of organisms in various taxonomic groups.[5,6] The autofluorescence of *S. mansoni* eggs in murine and human tissues (*e.g.*, liver and gut) and eggshells released by or located in the reproductive organs of *in vitro*-cultivated worms have been reported[7,8,9,10,11] (Table). In all these reports, the eggshell is identified as the source of fluorescence, although compelling evidence is lacking and some images do not clearly show fluorescent structures. In addition, these studies did not focus on the examination of parasite structure autofluorescence, which was reported as an undesired side effect in some cases.[12,13]

Sites of structural protein cross-linkage have been considered to be the main sites of autofluorescence in trematode eggshells.[14] We observed prominent autofluorescence signals from *S. mansoni* egg miracidia, suggesting that metabolically active or intact miracidia are the main contributors (Fig. 2). Contrary to our findings, most reports describe autofluorescence only in *S. mansoni* eggshells, which may be explained by the performance of procedures with whole worms or tissues that remove miracidia contents or prevent their autofluorescence. The similarity of our findings for formaldehyde-fixed eggs from MLs and alcohol-fixed eggs from HF does not support the potential for the degradation or removal of the molecular source of autofluorescence by fixatives or reduced pH.[15]

A distinctive noise-to-signal ratio is important when using autofluorescence to detect infectious agents, as the present results demonstrate.[13] The distinct fluorescence signature of the miracidium, especially under the B-2A filter, is a reliable marker that stands out among fecal debris. Such markers are essential for the development of automated detection systems, but they may also improve detection by conventional microscopy. Autofluorescence is not commonly used to detect parasites amid very diverse fecal debris. Sakurai and collaborators[16] submitted a patent for a procedure involving the use of autofluorescence to detect *Schistosoma japonicum* eggs.

In conclusion, we report that *S. mansoni* eggs autofluorescence in fecal debris, enabling their relatively clear distinction from the background. We provide compelling evidence that this autofluorescence originates mainly from miracidia, and not eggshells, contrasting with most reports in the literature. These findings sup-

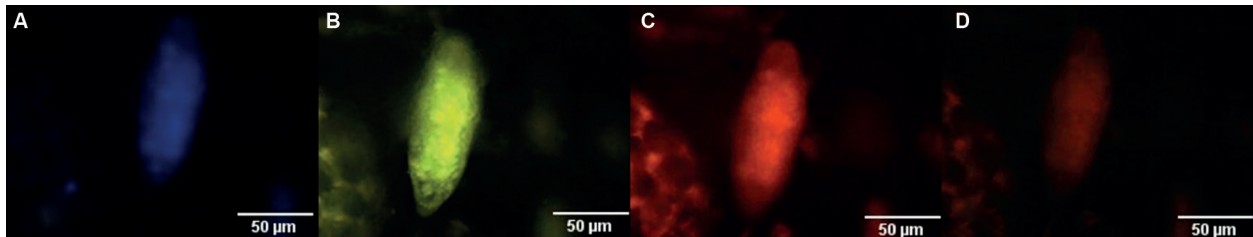

Fig. 1: autofluorescence of *Schistosoma mansoni* eggs from human fecal samples under microscope filters: (A) UV-2E (330-380 nm), Dichroic Mirror (DM) 400 nm, emission: > 420 longpass (LP), Blue; (B) B-2A (FITC, 420-495 nm), DM 505 nm, emission: > 515 LP, Green/Yellow; (C) G-2A(TRITC, 510-560 nm), 575 nm, emission: > 590 LP, Orange/Red; and (D) Y-2E/C (540-580 nm), 595 nm, emission: > 630 LP, Red.

TABLE

Reports from the literature describing autofluorescent *Schistosoma* spp eggs

| Reference | Egg sources | | | Main fluorescent structure |
| | Mice tissue | *In vitro* culture | Human tissue | |
| --- | --- | --- | --- | --- |
| Domingo 1968 | rectum / liver | | rectum | eggshell |
| Edwards 2015 | | inside the worm | | eggshell / miracidia (?) |
| Wang 2018 | | released by worms | | eggshell |
| Knhur 2018 | gut / liver | | | eggshell |
| Peterkova 2024 | liver | | | eggshell |

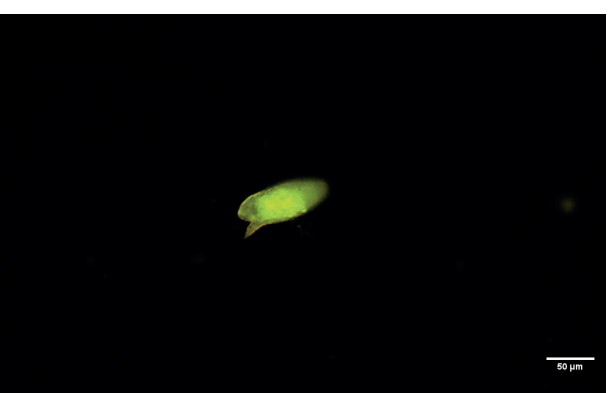

Fig. 2: *Schistosoma mansoni* egg obtained from the liver of an experimentally infected mouse under B-2A (FITC) filter (420-495 nm, blue; emission: > 515 nm longpass) showing miracidium as the source of a more intense autofluorescence.

port the hypothesis that functional, rather than structural, molecules are the primary source of autofluorescence in mature *S. mansoni* eggs.

## AUTHORS' CONTRIBUTION

DSCAP - conceptualization, data curation, formal analysis, investigation, methodology, visualization, writing - original draft, writing - review and editing; LOV - formal analysis, investigation, methodology, visualization; FTF - data curation, formal analysis, investigation, methodology; TH - conceptualization, formal analysis, investigation, methodology, visualization, writing - original draft, writing - review and editing; RP - conceptualization; MJ - conceptualization, writing - review and editing; SD - resources, writing - review and editing; DNC - resources, writing - review and editing; CG-T - conceptualization, formal analysis, investigation, methodology, visualization, project administration, supervision, writing - original draft, writing - review and editing. Funding sources had no role in the design, analysis, or writing of this manuscript.

## DATA AVAILABILITY

The contents underlying the research text are included in the manuscript.

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

# OPEN PEER REVIEW

Memórias do IOC thanks the anonymous reviewers for their contribution to the peer review of this work.

**FIRST REVIEW ROUND**

REVIEWERS' COMMENTS

**REVIEWER #1**

GENERAL COMMENTS
Despite their low sensitivity, especially for low-intensity infections, traditional egg-based parasitological tests are the most common method for diagnosing schistosomiasis mansoni. Therefore, there is an urgent need to test diagnostic methods that can overcome these limitations. One such alternative is Helminthex. Although it is more sensitive than current methods, it is laborious. Fluorescence microscopy, which takes advantage of the organism's autofluorescent properties, has also been a useful diagnostic tool in clinical parasitology. This study aims to improve the Helminthex technique by investigating the autofluorescence of Schistosoma mansoni in samples obtained from infected mouse livers and human feces. The main finding is that autofluorescence is present in miracidium, contrary to previous studies that found autofluorescence only in eggshell. Given the limited number of publications, this study is scientifically sound and offers promising advancements for its application to S. mansoni.

ABSTRACT SECTION
The abstract highlights the main issue to be investigated: one of the steps in improving the diagnostic technique for S. mansoni. The authors describe the methodology, key findings, and conclusions. "These findings may help to improve diagnostic methods, especially with automated egg detection systems". I agree.
" and contribute to the elimination of schistosomiasis as a public health problem". A set of measures, including access to safe water and adequate sanitation, as well as the treatment of infected individuals with praziquantel, are considered important components of schistosomiasis control. Therefore, the authors could conclude that laboratory diagnosis is a key measure that can help eliminate schistosomiasis as a public health problem.

MATERIAL AND METHODS SECTION
Line 57 - S. mansoni eggs were obtained from experimentally infected mouse livers.
The authors could report on the host strain and the week or stage of infection at which the livers were collected?
Line 68 – Typing error: 14, and 14 eggs,

RESULTS SECTION
FIGURE LEGENDS
Line 189- emission long: >515 nm?
Are there any reasons for using the terms emission length and emission long?

DISCUSSION SECTION
Line 114 – viable egg - What is the role of 10% formalin? At this stage of the experiment, there is no indication of activity (movement, for example) of miracidium. The term mature egg may be more accurate.

REFERENCES SECTION
Line 159 - Souza, R. P. et al. - out of alphabetical order.

AUTHORS' RESPONSE TO THE REVIEWERS

Responses to Reviewer: 1
Comment 1 - ABSTRACT SECTION
The abstract highlights the main issue to be investigated: one of the steps in improving the diagnostic technique for S. mansoni. The authors describe the methodology, key findings, and conclusions. "These findings may help to improve diagnostic methods, especially with automated egg detection systems". I agree.
" and contribute to the elimination of schistosomiasis as a public health problem". A set of measures, including access to safe water and adequate sanitation, as well as the treatment of infected individuals with praziquantel, are considered important components of schistosomiasis control. Therefore, the authors could conclude that laboratory diagnosis is a key measure that can help eliminate schistosomiasis as a public health problem.
Response to comment 1 – Agreed. The concluding sentence was changed to "Besides access to safe water and adequate sanitation, as well as health education and the treatment of infected individuals, laboratory diagnosis is a key measure that can help eliminate schistosomiasis as a public health problem." (highlighted in the new version of the manuscript)

Comment 2 - MATERIAL AND METHODS SECTION

Line 57 - S. mansoni eggs were obtained from experimentally infected mouse livers.

The authors could report on the host strain and the week or stage of infection at which the livers were collected?

Response to comment 2 – We now report: Line 59, "BALB/c" and Line 62, "Eggs isolated from ML after 7-8 weeks post infection"

Comment 3 - Line 68 – Typing error: 14, and 14 eggs,

Response to comment 3 – Actually it is not a typing error, but a lack of clarity: there were 3 sets of HF eggs: set 1 containing 10 eggs, set 2 containing 14 eggs and set 3 also containing 14 eggs. The phrase was rewritten: "The same procedure was performed with three sets containing 12 to 14 HF eggs".

Comment 4 - RESULTS SECTION, FIGURE LEGENDS

Line 189- emission long: >515 nm?

Are there any reasons for using the terms emission length and emission long?

Response to comment 4 – Agreed with the reviewer, the correct technical expression is "longpass" or abbreviated "LP" (both "long" and "length" are incorrect). Both legends were corrected.

Comment 5 - DISCUSSION SECTION

Line 114 – viable egg - What is the role of 10% formalin? At this stage of the experiment, there is no indication of activity (movement, for example) of miracidium. The term mature egg may be more accurate.

Response to comment 5 – Agreed, "viable" was removed and "mature" was introduced.

Comment 6 - REFERENCES SECTION

Line 159 - Souza, R. P. et al. - out of alphabetical order.

Response to comment 6 – Corrected, thank you.

## SECOND REVIEW ROUND

REVIEWERS' COMMENTS

**REVIEWER #1**

Despite their low sensitivity, especially for low-intensity infections, traditional egg-based parasitological tests are the most common method for diagnosing schistosomiasis mansoni. Therefore, there is an urgent need to test diagnostic methods that can overcome these limitations. One such alternative is Helminthex. Although it is more sensitive than current methods, it is laborious. Fluorescence microscopy, which takes advantage of the organism's autofluorescent properties, has also been a useful diagnostic tool in clinical parasitology. This study aims to improve the Helminthex technique by investigating the autofluorescence of Schistosoma mansoni in samples obtained from infected mouse livers and human feces. The main finding is that autofluorescence is present in miracidium, contrary to previous studies that found autofluorescence only in eggshell. Given the limited number of publications, this study is scientifically sound and offers promising advancements for its application to S. mansoni.

The authors addressed the above observations.

