## [Reviewer Report · FIRST REVIEW ROUND - REVIEWERS COMMENTS]

## REVIEWER #1

GENERAL COMMENTS

Despite their low sensitivity, especially for low-intensity infections, traditional egg-based parasitological tests are the most common method for diagnosing schistosomiasis mansoni. Therefore, there is an urgent need to test diagnostic methods that can overcome these limitations. One such alternative is Helminthex. Although it is more sensitive than current methods, it is laborious. Fluorescence microscopy, which takes advantage of the organism’s autofluorescent properties, has also been a useful diagnostic tool in clinical parasitology. This study aims to improve the Helminthex technique by investigating the autofluorescence of *Schistosoma mansoni* in samples obtained from infected mouse livers and human feces. The main finding is that autofluorescence is present in miracidium, contrary to previous studies that found autofluorescence only in eggshell. Given the limited number of publications, this study is scientifically sound and offers promising advancements for its application to *S. mansoni*.

## ABSTRACT SECTION

The abstract highlights the main issue to be investigated: one of the steps in improving the diagnostic technique for *S. mansoni*. The authors describe the methodology, key findings, and conclusions. “These findings may help to improve diagnostic methods, especially with automated egg detection systems”. I agree.

“ and contribute to the elimination of schistosomiasis as a public health problem”. A set of measures, including access to safe water and adequate sanitation, as well as the treatment of infected individuals with praziquantel, are considered important components of schistosomiasis control. Therefore, the authors could conclude that laboratory diagnosis is a key measure that can help eliminate schistosomiasis as a public health problem.

## MATERIAL AND METHODS SECTION

Line 57 - *S. mansoni* eggs were obtained from experimentally infected mouse livers. The authors could report on the host strain and the week or stage of infection at which the livers were collected?

Line 68 – Typing error: 14, and 14 eggs,

## RESULTS SECTION

FIGURE LEGENDS

Line 189- emission long: >515 nm? Are there any reasons for using the terms emission length and emission long?

## DISCUSSION SECTION

Line 114 – viable egg - What is the role of 10% formalin? At this stage of the experiment, there is no indication of activity (movement, for example) of miracidium. The term mature egg may be more accurate.

## REFERENCES SECTION

Line 159 - Souza, R. P. et al. - out of alphabetical order.

## AUTHORS’ RESPONSE TO THE REVIEWERS

Responses to Reviewer: 1

**Comment 1 - ABSTRACT SECTION**

Response to comment 1 – Agreed. The concluding sentence was changed to “Besides access to safe water and adequate sanitation, as well as health education and the treatment of infected individuals, laboratory diagnosis is a key measure that can help eliminate schistosomiasis as a public health problem.” (highlighted in the new version of the manuscript)

**Comment 2 - MATERIAL AND METHODS SECTION**

Response to comment 2 – We now report: Line 59, “BALB/c” and Line 62, “Eggs isolated from ML after 7-8 weeks post infection”

**Comment 3**

Response to comment 3 – Actually it is not a typing error, but a lack of clarity: there were 3 sets of HF eggs: set 1 containing 10 eggs, set 2 containing 14 eggs and set 3 also containing 14 eggs. The phrase was rewritten: “The same procedure was performed with three sets containing 12 to 14 HF eggs”.

**Comment 4 - RESULTS SECTION, FIGURE LEGENDS**

Response to comment 4 – Agreed with the reviewer, the correct technical expression is “longpass” or abbreviated “LP” (both “long” and “length” are incorrect). Both legends were corrected.

**Comment 5 - DISCUSSION SECTION**

Response to comment 5 – Agreed, “viable” was removed and “mature” was introduced.

**Comment 6 - REFERENCES SECTION**

Response to comment 6 – Corrected, thank you.

---

## [Reviewer Report · REVIEWERS COMMENTS]

## REVIEWER #1

Despite their low sensitivity, especially for low-intensity infections, traditional egg-based parasitological tests are the most common method for diagnosing schistosomiasis mansoni. Therefore, there is an urgent need to test diagnostic methods that can overcome these limitations. One such alternative is Helminthex. Although it is more sensitive than current methods, it is laborious. Fluorescence microscopy, which takes advantage of the organism’s autofluorescent properties, has also been a useful diagnostic tool in clinical parasitology. This study aims to improve the Helminthex technique by investigating the autofluorescence of *Schistosoma mansoni* in samples obtained from infected mouse livers and human feces. The main finding is that autofluorescence is present in miracidium, contrary to previous studies that found autofluorescence only in eggshell. Given the limited number of publications, this study is scientifically sound and offers promising advancements for its application to *S. mansoni*.

The authors addressed the above observations.